# Evaluation of Machine Learning Techniques for Traffic Flow-Based Intrusion Detection

**DOI:** 10.3390/s22239326

**Published:** 2022-11-30

**Authors:** María Rodríguez, Álvaro Alesanco, Lorena Mehavilla, José García

**Affiliations:** Aragón Institute of Engineering Research (I3A), University of Zaragoza, 50018 Zaragoza, Spain

**Keywords:** CICIDS2017, datasets, intrusion detection, machine learning, traffic flows, Weka, Zeek

## Abstract

Cybersecurity is one of the great challenges of today’s world. Rapid technological development has allowed society to prosper and improve the quality of life and the world is more dependent on new technologies. Managing security risks quickly and effectively, preventing, identifying, or mitigating them is a great challenge. The appearance of new attacks, and with more frequency, requires a constant update of threat detection methods. Traditional signature-based techniques are effective for known attacks, but they are not able to detect a new attack. For this reason, intrusion detection systems (IDS) that apply machine learning (ML) techniques represent an alternative that is gaining importance today. In this work, we have analyzed different machine learning techniques to determine which ones permit to obtain the best traffic classification results based on classification performance measurements and execution times, which is decisive for further real-time deployments. The CICIDS2017 dataset was selected in this work since it contains bidirectional traffic flows (derived from traffic captures) that include benign traffic and different types of up-to-date attacks. Each traffic flow is characterized by a set of connection-related attributes that can be used to model the traffic and distinguish between attacks and normal flows. The CICIDS2017 also contains the raw network traffic captures collected during the dataset creation in a packet-based format, thus permitting to extract the traffic flows from them. Various classification techniques have been evaluated using the Weka software: naive Bayes, logistic, multilayer perceptron, sequential minimal optimization, k-nearest neighbors, adaptive boosting, OneR, J48, PART, and random forest. As a general result, methods based on decision trees (PART, J48, and random forest) have turned out to be the most efficient with F1 values above 0.999 (average obtained in the complete dataset). Moreover, multiclass classification (distinguishing between different types of attack) and binary classification (distinguishing only between normal traffic and attack) have been compared, and the effect of reducing the number of attributes using the correlation-based feature selection (CFS) technique has been evaluated. By reducing the complexity in binary classification, better results can be obtained, and by selecting a reduced set of the most relevant attributes, less time is required (above 30% of decrease in the time required to test the model) at the cost of a small performance loss. The tree-based techniques with CFS attribute selection (six attributes selected) reached F1 values above 0.990 in the complete dataset. Finally, a conventional tool like Zeek has been used to process the raw traffic captures to identify the traffic flows and to obtain a reduced set of attributes from these flows. The classification results obtained using tree-based techniques (with 14 Zeek-based attributes) were also very high, with F1 above 0.997 (average obtained in the complete dataset) and low execution times (allowing several hundred thousand flows/s to be processed). These classification results obtained on the CICIDS2017 dataset allow us to affirm that the tree-based machine learning techniques may be appropriate in the flow-based intrusion detection problem and that algorithms, such as PART or J48, may offer a faster alternative solution to the RF technique.

## 1. Introduction

The increase of cyberattacks threatening institutions, enterprises, and end users places them on the verge of disaster more frequently than imagined. According to Check Point Research [1], global attacks increased by 28% in the third quarter of 2022 compared to same period in 2021. The average weekly attacks per organization worldwide reached over 1130. Although continuous advances in both technology and secure practices try to reduce these risks to its minimum expression, an absolutely secure system does not exist. Cybersecurity incidents are always present. Thus, cybersecurity practices and procedures must be implemented to minimize them. In this strategy, network security is crucial. Network security encompasses many essential aspects of a successful cybersecurity strategy, from secure firewall configuring to secure device management, touching VPN configuration, VLAN segmentation, etc. One of the most interesting fields in network security is intrusion detection because it can be used both for detecting ongoing attacks to network or end devices and for finding out already compromised devices. Therefore, intrusion detection systems (IDS) emerged in order to detect malicious traffic and compromised networks and systems. IDSs fetch the data, preprocess it, and finally determine whether or not an event is considered malicious [2,3,4]. The Intrusion Detection Working Group (IDWG), formerly known as Common Intrusion Detection Framework (CIDF), is a group created by DARPA, later integrated into the IETF, which defined an IDS architecture. It considers four types of modules: E blocks or event-boxes, which through sensors monitor the system to obtain information on events to be analyzed by other blocks; D blocks or database-boxes, which store information from E blocks to be processed by A and R boxes; A blocks or analysis-boxes, which are processing modules that analyze events and detect possible hostile behavior; and R blocks or response-boxes, which execute a response in case of intrusion in order to thwart the threat. Depending on the information source (E blocks), two types of IDS are distinguished: host-based IDS and network-based IDS. Network-based IDSs monitor events, such as flows or firewall logs, analyzing the traffic traversing the network and verifying its content. On the other hand, host-based IDSs work with system-related events, e.g., syslog, file systems, CPU load, etc., to detect threats inside these end hosts. Another classification can be made based on the analysis used (A blocks). Signature-based IDSs search for defined patterns (signatures) generated from previous detected intrusions added to a list, and thus allow to detect an attack whose pattern is already stored, although it is more difficult to detect new attacks. On the other hand, anomaly-based IDS (A-IDS) build a model of normal behavior from training data so that any data that do not conform to that model is considered anomalous [4]. Depending on the type of processing applied in relation to the “behavior” model, anomaly detection techniques can be classified into three categories: statistically based (the behavior of the system is modeled from a statistical point of view), knowledge based (modeling the behavior from available data such as protocol specifications, traffic instances, etc.), and machine learning (ML) based, which rely on the establishment of an explicit or implicit model [3,4].

Different network attacks can have different identification patterns, corresponding to a set of events that can compromise network security if left undetected. Several characteristics can be extracted from network traffic to model those patterns, such as packet sizes, durations, protocols, services, and flags. These patterns are derived from network traffic data, hence the importance of data collection in order to further apply ML techniques. Network traffic packets can be analyzed independently, taking advantage of techniques, such as deep packet inspection (DPI) [5], or by grouping them into traffic flows, where the main characteristics of a communication between peers are extracted. A flow is defined by its 5-tuple, a collection of five data points: the source and destination IP addresses exchanging information, the source and destination ports if any, and the transport layer protocol. Flow identifies a communication channel, and all packets sharing the same 5-tuple fields belong to the same flow. IDS that implement DPI techniques to detect anomalies or attacks rely on a thoughtful packet content revision. If DPI is required, packets must be analyzed individually. DPI techniques are very resource consuming and their application is limited to unencrypted traffic. Nowadays, firewalls can implement secure sockets layer (SSL) inspection to decrypt transport layer security (TLS) traffic, but this technique is very intrusive and limited to network boundaries, where firewalls are placed. Traffic flows used for IDS are vectors of data characterizing the flow, and hence individual packet content is not stored. IDS that use traffic flows are gaining much attention because they can be more resource-efficient, and because they fit well to ML techniques due to the vectorized nature of the traffic flow [6]. Besides, widely implemented protocols in network devices such as Netflow [7] and popular IDS such as Zeek [8] already produce this kind of traffic flow information.

Recently, new datasets made up of benign traffic and varied types of attacks have been collected, which serve as test scenarios for IDS evaluation. The databases contain labeled packets or network flows that are made up of certain features extracted from network traffic. Some very complete reviews of the work carried out in the application of ML techniques over different datasets for intrusion detection can be found in [4,9,10,11,12,13,14,15].

The main objective of this work is the application and evaluation of classification techniques that allow distinguishing between normal traffic flows and traffic flows corresponding to attacks over the CICIDS2017. The main contributions to knowledge produced during this research work are highlighted below:An extensive benchmark study on the different machine learning techniques applied over the CICIDS2017 dataset to determine which ones better perform considering both classification scores and execution times.Evaluate the differences between multiclass classification (distinguishing between different types of attack) and binary classification (distinguishing only between normal traffic and attack flows).Evaluate the use of a correlation-based feature selection (CFS) technique for traffic flow attributes.Evaluate the use of Zeek to extract the traffic flows and attributes and compare the classification results obtained with respect to those obtained using the original CICIDS2017 dataset flows and attributes.

The remainder of the paper is structured as follows. Section 2 provides a description of the main materials and methods used in this work, including an overview of the available datasets and a complete description of the selected dataset, and the machine learning techniques applied. Section 3 presents the methodology followed for traffic flow classification considering different approaches to the problem. The evaluation and performance results of the techniques are presented in Section 4. The discussion and finally the conclusions are presented in Section 5 and Section 6, respectively.

## 2. Materials and Methods

### 2.1. Intrusion Detection Datasets

As has been said, IT security has become a crucial issue and much effort has been directed to the research of intrusion and anomaly traffic detection. Many contributions have been published in the analysis of security-related data, which usually require network-based datasets to assess the methods. However, the lack of representative publicly available datasets constitutes one of the main challenges for anomaly-based intrusion detection [16]. A relevant aspect to establish a basic dataset taxonomy regards to the format of the network traffic collected in the dataset. It can be captured either in packet-based or flow-based format. Capturing network traffic at packet-level is usually done by mirroring ports on a network device (commonly captured in .pcap format) and packet-based data include complete payload information. On the other hand, a more recent approach is based on flow-based data, which represents traffic in an aggregated way and usually contains only metadata or derived features from the network connections. Flow-based data aggregate all packets which share some properties within a time window into one flow (in unidirectional or bidirectional format) and usually do not include any payload. Sometimes, a third category with no standard format has been introduced in some works, which could include flow-based datasets enriched with additional information from packet-based data or host-based log files. In addition to this basic format feature, other relevant properties have been identified and proposed in [16] to assess the suitability of the datasets, which can be grouped into five categories: general information, nature of the data, data volume, recording environment and evaluation. A comprehensive review of IDS datasets can be found in [16,17] and some of the most relevant are briefly described below.

DARPA [18,19]. DARPA 1998/99 datasets, created at the MIT Lincoln Lab within an emulated network environment, are probably the most popular data sets for intrusion detection. DARPA 1998 and DARPA 1999 datasets contain seven and five weeks, respectively, of network traffic in packet-based format, including different types of attacks, e.g., DoS, port scans, buffer overflow, or rootkits. These datasets have been criticized due to artificial attack injections or redundancies [20,21].NSL-KDD [21]. It is an improved version of KDDCUP99 (which is based on the DARPA dataset) developed in the University of California. To avoid the large amount of redundancy in KDDCUP99, the authors of NSL-KDD removed duplicates from the KDDCUP99 dataset and created more sophisticated predefined subsets divided into training and test subsets. NSL-KDD uses the same attributes as KDDCUP99 and belongs to the third format category described before. The underlying network traffic of NSL-KDD dates back to the year 1998 but it has been a reference until today.ISCX 2012 [22]. This dataset was created in the Canadian Institute for Cybersecurity by capturing traffic in an emulated network environment launched during one week. A systematic dynamic approach was proposed to generate an intrusion detection dataset including normal and malicious network behavior through the definition of different profiles. These profiles determine attack scenarios, such as SSH brute force, DoS, or DDoS, and normal user behavior, e.g., writing e-mails or browsing the web. The profiles were used to create the dataset both in packet-based and bidirectional flow-based format and its dynamic approach allows generation of new datasets.UGR’16 [23]. This dataset was created at the University of Granada and it is a unidirectional flow-based dataset, which relies on capturing periodic effects in an ISP environment along four months. IP addresses were anonymized, and the flows were labeled as normal, background, or attack. The attacks correspond either to explicitly performed attacks (botnet, DoS, and port scans) or to manually identified and labeled as attacks. Most of the traffic is labeled as background which could be normal or an attack.UNSW-NB15 [24]. This dataset created by the Australian Centre for Cyber Security includes normal and malicious network traffic in packet-based format captured in an emulated environment The dataset is also available in flow-based format with additional attributes. It contains nine different families of attacks, e.g., backdoors, DoS, exploits, fuzzers, or worms. UNSW-NB15 comes along with predefined splits for training and test.CICIDS2017 [25]. This dataset was created at the Canadian Institute for Cybersecurity within an emulated environment over a period of 5 days and contains network traffic in packet-based and bidirectional flow-based format (each flow characterized with more than 80 attributes). Normal user behavior is executed through scripts. The dataset contains a wide range of attack types, such as SSH brute force, heartbleed, botnet, DoS, DDoS, web and infiltration attacks. CICIDS2017 has been selected in this work due to its novelty and the properties considered in its development. It is described in more detail below.

Besides network-based datasets, other data sources, such as data repositories (CAIDA, AZSecure, DEF CON CTF Archive, Internet Traffic Archive, Kaggle, Malware Traffic Analysis, MAWILab, etc.) and traffic generators, can be helpful in IDS evaluation.

### 2.2. CICIDS2017 Dataset

The CICIDS2017 dataset was developed by the Canadian Institute of Cyber Security, by performing intrusion traffic characterization [25]. This dataset consists of seven attack categories that have been applied in an attack scenario. The testbed infrastructure was divided into two completely separated networks, denoted as attack-network and victim-network, where common communication devices were included (router, firewall and switch) along with the different versions of the common three operating systems (Windows, Linux, and Macintosh). This dataset was developed taking the following characteristics into account: diversity of attacks, anonymity, available protocols, capturing the complete network traffic, capturing the complete network interaction, defining the complete configuration of the network, feature set, labeled data samples, heterogeneity, and metadata [26]. In contrast to KDDCUP99 and NSL-KDD datasets which considered a few attack categories, CICIDS2017 dataset includes a wider range of attacks, such as distributed denial of service, denial of service, brute force, XSS, SQL injection, botnet, web attack, and infiltration (see Table 1 and Table 2 for a complete description of files and flows). The CICIDS2017 dataset includes 2,830,540 flows labeled using the CICFlowMeter tool [27] with more than 80 features in each flow. Therefore, it is a high dimensionality, multi-class, and high-class imbalanced dataset. It also includes the network traffic in packet-based format for a total of 11,522,402 packets.

The main characteristics of the different attacks are described below and with more detail in [25,26].

Brute force attack: it is basically a hit and try attack, which can be used for password cracking and also to discover hidden pages and content in a web application.Heartbleed attack: it is performed by sending a malformed heartbeat request with a small payload and large length field to the vulnerable party (e.g., a server) in order to trigger the victim’s response. It comes from a bug in the OpenSSL cryptography library of the transport layer security (TLS) protocol.Botnet: Networks of hijacked computers (named zombies) under cybercriminal control used to carry out various scams, spam campaigns and cyberattacks. Zombie computers communicates with a command-and-control center to receive attack instructions.DoS Attack: The attacker tries to make the machine or network resource temporally unavailable. It is usually achieved by flooding the target machine or resource with superfluous requests in an attempt to overload systems and prevent some or all legitimate requests from being fulfilled.DDoS attack: The result of multiple compromised systems that flood the targeted system by generating large network traffic, overflowing the bandwidth or resources of the victim.Web attack: Targets vulnerabilities in websites to gain unauthorized access, obtain confidential information, introduce malicious content, or alter the website’s content. In SQL injection, the attacker uses a string of SQL commands to force the database to reply. In cross-site scripting (XSS), attackers find the possibility of script injection when the code is not properly tested, and in Brute Force over HTTP they try a list of passwords to find the administrator’s password.Infiltration attack: It is usually applied by exploiting vulnerable software in the target’s computer. After successful exploitation, a backdoor is executed on the victim’s computer which can perform other attacks on the network (full port scan, service detection, etc.).

### 2.3. Machine Learning Techniques for Classification

The techniques used in the classification of traffic flows are described now in more detail. First, the basics of the operation of classification techniques based on supervised learning are presented, and then the main indicators used to evaluate the performance of the classifiers are shown. In general, it can be stated that a classification algorithm based on supervised learning aims to extract knowledge from a data set (training set) and to model that knowledge for its subsequent application in decision making on a new data set (test set). Mathematically, supervised learning starts with a data set that is a collection of labeled examples {(x_i_, y_i_)} with i = 1 … N, where each element x_i_ is called a feature vector, where each dimension j = 1, …, D contains a value that describes the example in some way. This value is called characteristic or attribute and is denoted as x_i_(j). For all instances in the data set, the feature at position j in the feature vector always contains the same type of information (e.g., the number of packets in a traffic flow). The label (label) denoted as y_i_ can be an element belonging to a finite set of classes {1, 2, …, C}. The class can be viewed as a category to which an instance belongs (e.g., the class that represents a specific type of attack). The objective of a supervised learning algorithm is to use the data set {(x_i_, y_i_)} to produce a classification model that allows taking a new feature vector x as input information and provides with an output of the label that should be assigned to that vector [28]. The most commonly used families of classifiers are presented below, along with a brief description of the operation of the ML classifiers used in this work.

**Bayesian Classifiers:** Bayesian methods quantitatively provide a probabilistic measure of the relevance of the variables in the classification problem. In the application of these methods, correlation between the attributes of the training set must be avoided, since this would invalidate their results.

Naive Bayes: This classifier is a probabilistic method based on the computation of conditional probabilities and Bayes’ theorem. It is described as naive due to the simplifications that determine the independence hypothesis of the predictor variables, i.e., naive Bayes starts from the hypothesis that all the attributes are independent of each other [28,29].

**Functions:** In this group of methods, we have included those that generate a classification function and that do not explicitly obtain a tree or set of rules.

Logistic: In the multinomial logistic regression model, the logistic regression method is generalized for multiclass problems and is therefore used to predict the probabilities of the different possible outcomes of a categorical distribution as a dependent variable (class), from of a set of independent variables (attributes) [28].Multi-layer perceptron (MLP): This is an artificial neural network made up of multiple layers, which allows solving problems that are not linearly separable [28]. In the network used, three layers have been considered: an input layer where the values of the attributes are introduced, a hidden layer to which all the input nodes are connected; and an output layer, in which the classification values of the instances are obtained according to the classes.Sequential minimal optimization (SMO): This is a classifier in which John Platt’s sequential minimum optimization algorithm [30] is used in order to train a support vector machine classifier (SVM).

**Instance-Based Learning:** In this type of learning, the training examples are stored and to classify a new instance the most similar previously classified instances are extracted and their classification is used to classify the new one. In these techniques, the learning process is trivial (lazy learners) and the classification process is the most time consuming.

k-nearest neighbors (IBk): The k-NN method is based on classifying an instance according to the classes of the k-most similar training instances. It is a non-parametric classification method that allows estimating the probability density function or directly the a posteriori probability that an instance belongs to a class using the information provided by the set of classified instances [28,31].

**Metaclassifiers:** This family would include those complex classifiers that are either obtained through the composition of simple classifiers or include some preprocessing of the data used.

Adaptive boosting (AB): This is a meta-algorithm for statistical classification that can be used in combination with other learning algorithms to improve the performance [28,32]. In this way, the output of the other learning algorithms (weak learners) is combined into a weighted sum that represents the final output of the non-linear classifier.

**Rule based classifiers:** These are methods or procedures that allow the generation of classification rules (coverage rule learning) [28]. In these algorithms, a rule or condition whose fulfillment is covered by the largest possible number of instances of one of the classes is sought, a process that is repeated iteratively until the classification is finished. Among these algorithms, the simplest are ZeroR, in which any instance is classified as belonging to the most frequent class, and OneR (1R), which searches for a single rule on the most discriminating attribute.

OneR: This is one of the simplest and fastest classification algorithms, although sometimes its results are surprisingly good compared to much more complex algorithms [33]. In OneR, a rule is generated for each attribute and the one that provides with the least error in the classification is selected.

**Decision trees:** A decision tree is a classifier that tries to find the best option at each step or decision that is made in the tree, so that each selected partition maximizes some discrimination criterion (classification error, entropy gain, etc.) [28]. Trees have the advantage of providing an intuitive way to visualize the classification of a dataset.

J48 algorithm: This algorithm is an implementation of C4.5 [34], one of the most widely used in many data mining applications, and is based on the generation of a decision tree from of the data by partitioning recursively. The procedure to generate the decision tree consists in selecting one of the attributes as the root of the tree and creating a branch with each of the possible values of that attribute. In each resulting branch, a new node is created, and the same process is repeated, i.e., another attribute is selected and a new branch is generated for each possible value of it. The process is repeated until all instances have been sorted through some path in the tree.PART: This algorithm constitutes a simplified method (partial) of the C4.5 algorithm and is based on the construction of rules [35]. In this algorithm, the branches of the decision tree are replaced by rules, and although the underlying operation is the same, in this case only the branches of the tree that are most effective in selecting a class are included, which facilitates the task of programming the classifier. In each iteration, a partial pruned C4.5 decision tree is built, and the best leaf obtained (the one that allows classifying the largest number of examples) is transformed into a decision rule; then the created tree is deleted. Each time a rule is generated, the instances covered by that rule are removed and rules continue to be generated until there are no instances left to classify. The strategy used allows for great flexibility and speed. Using PART does not generate a complete tree, but rather a partial one in which the construction and pruning functions are combined until a stable subtree that cannot be simplified is found, at which point the rule is generated from that subtree.Random Forest (RF): In this technique, random forests are built by packing sets of random trees [28,36]. Trees built using the algorithm consider a certain number of random features at each node, without performing any pruning. The algorithm works by randomly testing a multitude of models, so that hundreds of decision trees can be combined, and then each decision tree is trained on a different selection of instances. The final random forest predictions are made by averaging the predictions obtained for each individual tree. Using random forest can reduce the overfitting effect of individual decision trees at the expense of increasing the computational complexity.

### 2.4. Computational Complexity

Although all the tests have been carried out in offline mode in this work, the final objective should be to have an IDS that can operate in real-time mode. When training classification methods aimed to work in real-time mode at least three factors should be considered: time complexity, incremental update capability, and generalization capacity [9]. The computational complexity for the algorithms included in this work are shown in Table 3 (corresponding to the model creation and test phases in columns two and three, respectively). It is assumed that the dataset consists of n instances (traffic flows) described by m attributes and that n >> m (other specific parameters of the ML techniques are also included in the table). Although some of the expressions obtained from the literature can be very dependent on the specific implementation carried out, these values permit to have an estimation of the time complexity. As a general rule, algorithms with complexity O(n) and O(nlog n) may be considered to work in real-time, and those with O(n2) would have an acceptable time complexity in many situations. On the other hand, it is usually considered that O(n3) and higher orders correspond to much slower algorithms and should only be used in offline approaches [9]. In the testing phase, the computational complexity is usually less and the execution times are fast, mostly on the order of linear time with respect to the input data size (except in lazy learner classifiers such as k-NN where most of the time is spent in the testing phase).

### 2.5. Performance Metrics

The confusion matrix (see Table 4) shows the number of flows correctly or incorrectly classified. The factors of the classification are: TP (true positives) is the number of correctly classified attacks, TN (true negatives) is the number of normal flows correctly classified, FP (false positives) is the number of normal instances misclassified as attacks, and FN (false negatives) is the number of attack instances misclassified as normal. Based on these elements, the following indicators can be defined [28].

**Accuracy**: is the ratio of correctly classified traffic flows to the total number of flows. Its complementary (1-Acc) is the error rate. It is frequently used to measure the effectiveness of the classification algorithms and it is also known as classification rate (CR). In imbalanced domains, instances of one class outnumber instances of other classes (for example, the number of normal traffic flows could be much higher than the number of attack flows). In these cases, accuracy can be misleading, and it is necessary to use other indicators, such as recall and precision.
(1)$$\mathrm{Accuracy}\;=\frac{\mathrm{TP}+\mathrm{TN}}{\mathrm{TP}+\mathrm{TN}+\mathrm{FP}+\mathrm{FN}}$$

**Recall** or **True Positive Rate (TPR)**: represents the probability that a positive example is correctly identified by the classifier and is also known as detection rate (DR) or sensitivity.
(2)$$\mathrm{TPR}\;=\frac{\mathrm{TP}}{\mathrm{TP}+\mathrm{FN}}$$

**Precision**: represents the ratio of true positives to all examples classified as positive. It is a measure of the estimated probability of a correct positive prediction and is also called the Positive Predictive Value (PPV). Note that when TP is 0 (no attack has been correctly classified) and FP is 0 (all benign flows have been correctly classified), a valid numerical result is not obtained (in this case, the “?” symbol will be indicated). While precision is the frequency of true positives (real attacks) among all examples considered positive by the classifier (flows classified as attacks), recall is the frequency of true positives (real attacks) among all positive examples in the set (attacks on the dataset).
(3)$$\mathrm{Precision}\;=\frac{\mathrm{TP}}{\mathrm{TP}+\mathrm{FP}}$$

**F-Measure** or **F1-Score**: F-Measure combines precision and recall in a single weighted indicator. In the case of assigning the same weight to both, F1 is derived.
(4)$$F1=\frac{2\;\mathrm{TP}}{2\;\mathrm{TP}+\mathrm{FP}+\mathrm{FN}}=2\frac{\mathrm{precision}\;\mathrm{recall}}{\mathrm{precision}+\mathrm{recall}}\;$$

In the field of traffic flow classification, a good classifier should have a high attack detection rate (TP) and a low false alarm rate (FP). Otherwise, if the system has a low detection rate, one will have a false sense of security, and if it has a high rate of false alarms, a lot of time will be spent analyzing those alarms. In this context, the base-rate fallacy states that if the number of attack flows is small compared to the number of benign traffic flows (which usually happens in the real world and is reflected in imbalanced datasets), the false alarm rate will be high unless the classifier has a very high recall or detection rate. When imbalanced sets are considered, some additional indicators, e.g., **Cohen’s kappa coefficient (κ)** or **Matthews correlation coefficient (MCC)** can be taken into account to estimate the improvement provided by the classifier with respect to the accuracy that would occur by mere chance. Cohen’s kappa coefficient is a statistic used to measure inter-rater reliability (and intra-rater reliability) for qualitative (categorical) items. It is generally seen as a more robust measure than simple percent agreement calculation, as κ takes into account the possibility of the agreement occurring by random classification. κ is defined as the ratio between what the accuracy has improved with the classification model used with respect to the maximum possible improvement. A value of κ close to 0 means that the classifier performs as a random classifier, i.e., the classification is mainly due to chance agreement whereas a value of κ close to 1 indicates that it performs perfectly by improving the results as much as it is possible due the discriminative power of the classification technique. On the other hand, Matthews correlation coefficient takes into account all four values in the confusion matrix, and a high value (MCC close to 1) means that both classes are well predicted, even if classes are imbalanced (no class is more important than the other). When the classifier always misclassifies (MCC = −1) represents perfect negative correlation, whereas MCC = 0 means that the classifier is no better than mere chance.

There are different validation methods that allow obtaining the described performance metrics and indicators. The most basic technique is known as simple validation or train-test, in which the model is created using the training set and applied to the test set. Different splits (split percentage) can be set to divide an original set into training and test subsets. On the other hand, in the cross-validation technique with n-folds, the dataset is divided into n equal-size disjoint subsets, so that the model is built over n-1 folds and tested in the remaining fold, a process that is repeated for all combinations which leads to the averaged metrics. However, in this context, it could be risky to evaluate the performance of the classification algorithms using 10-fold cross-validation (method used in some other works) taking nine out of 10 parts of the CICIDS2017 dataset to train the model, as it would be highly likely that traffic flows from the same attack would be found in both the training and test subsets. To reduce this possibility, in this work, we have considered it more appropriate to use a percentage split of the CICIDS2017 dataset using 50% for training and 50% for testing.

### 2.6. Testbed Framework

WEKA (Waikato Environment for Knowledge Analysis) [37,38] was selected for the application of ML techniques. Weka is a GNU-GPL licensed software developed by the University of Waikato (New Zealand) that consists of a set of JAVA libraries that allow extracting knowledge from databases. It was chosen due to its variety of algorithms, the presence of an intuitive graphical interface, and the possibility of carrying out data filtering, classification, and attribute selection in the same program. Weka has been run on a computer with the following specifications: Windows 10 Operating System, Inter Core i5-6600K 3.5 GHz 4 CPU processor, and 32 GB RAM. 20 GB of RAM was allocated exclusively for Weka.

## 3. Proposed Methodology

This section presents the methodology followed in the application of the ML techniques over the CICIDS2017 dataset. The general workflow is shown in Figure 1. We have followed two different approaches: either using the labelled traffic flows included in the dataset (.csv format files), where the attributes of each flow are available, or using the raw packet captures included in the dataset (.pcap format files), in which case it is necessary to detect and label the flows and to derive the flow attributes. Both approaches are described in more detail below.

### 3.1. Classification of Traffic Using the Traffic Flows in CICIDS2017 Dataset

This analysis corresponds to the application of ML techniques on the original .csv files that contain the attributes or features of the labelled traffic flows (files in .csv format). First, data cleaning was performed on the .csv format files in an error debugging process. Each of these files was transformed by means of a Python function to substitute with zero those values of numerical attributes that contained “Infinity”, “inf” or “NaN” in the dataset. A label was also added to the flow if it was not present. Moreover, Fwd Header Length attribute was removed because it was duplicated. From the original features included in the dataset, the Flow ID, Source and Destination IP, Source and Destination Port, and Timestamp were removed and not included in the classification, even prior to the feature selection process (leaving a total of 77 attributes for each flow). In contrast to other studies, Destination Port was also removed from the features set, since this attribute is part of the flow definition (every flow is characterized by its timestamp and source and destination IPs and ports). Therefore, no information regarding these attributes should be included to train and evaluate the ML techniques, since every specific attack was generated from the same IPs and ports, and then traffic flows could appear with identic values both in training and test subsets being easily detected as an attack. Removing these attributes should prevent a flow-based IDS from associating the timestamp or some host-specific information with a certain class without learning the underlying flow characteristics. Moreover, as many ports are dynamically used and some applications can be transmitted over the same port, it can be misleading to use this feature to detect attacks [39].

Then, the training and test subsets are defined considering a 50/50% split, instead of using the cross-validation method as argued above in Section 2.5. At this point, different alternatives for testing the ML techniques can be distinguished: the first one corresponds to a multiclass classification, and the second one corresponds to a binary classification. In multiclass classification, the different specific types of attacks included in each dataset file are distinguished as originally recorded in the CICIDS2017 dataset (see Table 1 and Table 2). On the other hand, a binary classification was carried out in which similar attacks present in each dataset file (e.g., different types of web attacks, different types of DoS, etc.) have been grouped using a Python script. Thus, the labels that distinguished them have been removed and a generic label indicating ATTACK has been included instead. Therefore, the classifier makes a binary distinction between BENING and ATTACK flows. As a last additional alternative, all the flows present on the different days were grouped in a single file (noted as all files) for the complete dataset.

Next, each classifier (naive Bayes, logistic, multilayer perceptron, SMO, Ibk, AB, OneR, PART, J48 and random forest) is applied independently over the defined subsets and considering the different alternatives for the classification (multiclass, binary, or binary using the complete dataset). The most widely used ML classifiers were tested in this work so that the comparison would be as broad as possible in line with previous research approaches [9,10,11,12,13,14,15,40]. The Weka parameter settings of the implemented ML methods in this work were the following: naive Bayes (NumDecimalPlaces = 2; useKernelEstimator = False), logistic (numDecimalPlaces = 4; ridge = 1.0 × 10^−8^; doNotStandardizeAttributes = False), multilayer perceptron (hiddenLayers = 1; learningRate = 0.3; validationThreshold = 20; trainingTime = 300 epochs), SMO (complexity parameter C = 1.0; Kernel = PolyKernel; calibrator = Logistic), Ibk (NumDecimalPlaces = 2; KNN = 1; nearestNeighbourSearchAlgorithm = LinearNNSearch), AB (classifier = DecisionStump; numIterations = 10; weightThreshold = 100), OneR (NumDecimalPlaces = 2; minBucketSize = 6), PART and J48 (confidenceFactor = 0.25; numFolds = 3; minNumObj = 2; useMDLcorrection = True), and random forest (numIterations = 100; maxDepth = unlimited; bagSizePercent = 100; numFeatures = int(log_2(#predictors)+1)).

Finally, the results for each classification technique and configuration are obtained and analyzed. Various performance metrics were collected, including accuracy, precision, recall, F1 score, Cohen’s kappa coefficient, MCC, ROC area, and execution times during training and testing phases. For the sake of simplicity and conciseness, we have mainly chosen the F1 score as the reference measure to analyze and compare the results.

### 3.2. Classification Using the Traffic Flows in CICIDS2017 Dataset and Attributes Selection

A prior selection of the most relevant attributes can be done before applying the ML techniques. Feature selection (FS) is a proven method for addressing the dimensionality problem [41,42,43,44] and the rationale for applying FS methods can be twofold. First, there are techniques that reduce their performance in the presence of non-relevant attributes, as is the case of the naive Bayes method (where redundant or correlated attributes violate the assumptions of applicability of the method). The number of instances for some attack classes are insufficient to justify the use of so many features for flow classification and indiscriminate use of all features makes it all easy to generate overfitted classifiers that are not sufficiently generic [14]. In addition, the greater the number of attributes, the greater the computational complexity and the execution time of the algorithms, so it may be convenient to limit the number of attributes used to a certain extent. Previous research has shown that irrelevant features, along with redundant features, can severely affect the accuracy of the classifiers [41,42,43,44,45,46]. Therefore, there is a need for finding an efficient feature selection method to identify and remove irrelevant and redundant information.

There are several techniques to carry out feature selection and some comprehensive reviews of them can be found in [41,42,43,44,45]. Filter feature ranking (FFR) methods consider the properties of a given dataset to evaluate and rank the features which are selected based on their rank scores [44]. The attributes are selected and evaluated independently of the learning algorithm used later in the classification. Some FFR methods are: information gain filter, gain ratio attribute filter, symmetrical uncertainty filter, probability significance filter, relief feature filter, weighted relief feature filter, chi-squared filter, correlation filter, etc. In filter-feature subset selection (FSS) methods, the features are evaluated and ranked by a search method which serves as the subset evaluator for the FSS methods. These search methods produce and evaluate features based on their usefulness towards better prediction performance. The most known FSS techniques are correlation-based feature subset selection (CFS) and consistency feature subset selection (CNS). On the other hand, the use of wrapper-based feature selection (WFS) methods considers a classifier to determine the suitability of a subset of attributes, so that it returns the most significant attributes for that classifier. The classifier is known in advance and the generated feature subsets are usually biased to the base classifier used for its evaluation [45].

In this study we have followed the filter-based approach since wrapper techniques require a long time to carry out the selection and because they require a complete training and evaluation process at each search step. This process becomes more complex if the number of instances is high. Feature selection involves searching through all possible combinations of attributes in the data to determine which subset of features ranks best. To do this, two elements are to be configured: an attribute evaluator and a search method. The evaluator determines which method is used to assign a value to each subset of attributes while the search method determines what type of search is performed. Hence, in order to carry out the selection of attributes, an attribute evaluator is first used to evaluate each of the cases and provide each attribute with a specific weight. The search method is in charge of generating the test space to find a good subset of attributes. In this work, the correlation-based feature selection (CFS) technique [47] was selected among others [42,44,45,46] as the attribute evaluator. CFS evaluates the value of a subset of attributes considering the individual classification capacity of each attribute together with the degree of redundancy between them. This method leads to selecting the subsets of characteristics that are highly correlated with the class and that have a low correlation with each other (those attributes that are redundant are eliminated) [47]. Therefore, the CFS method provides the best subset of features to operate together, not to be confused with the subset of the best features sorted or ranked by their own classification capacity (as, for example, a Ranker selection method would provide). CFS is a simple filter algorithm that ranks feature subsets according to the following correlation based heuristic evaluation function (see Equation (5)), where $M_{s}$ is the heuristic merit of a feature subset *S* containing *k* features, $\bar{r_{cf}}$ is the mean feature-class correlation (*f* Є *S*), and $\bar{r_{ff}}$ is the average feature-feature inter-correlation [47]. The numerator provides a measure of how predictive the class of a set of features is, while the denominator provides a measure of how much redundancy there is among the features.
(5)$$M_{s}=\frac{k\;\bar{r_{cf}}}{\sqrt{k+k\left(k-1\right)\bar{r_{ff}}}}$$

As search method, the well-known best first technique was used, which searches the space of attribute subsets by greedy hill climbing with an additional backtracking facility. Best first can start either with the empty set of attributes and search forward, or with the full set of attributes and search backward, or even start at any intermediate point and search in both directions. Therefore, the search progresses either forward through the search space by making local changes to the current feature subset adding single features or the search moves backward deleting single features. To prevent the method from exploring the entire feature subset search space in an exhaustive way, a stop criterion can be applied to end the search if several consecutive fully expanded subsets do not show any improvement over the best subset. Therefore, the search finishes automatically when the stop criterion is met and in each case it will end with a determined number of selected attributes. Therefore, the features subsets presented along the paper for every attack file or for the entire dataset correspond to the optimal selection according to the selected FS technique and the stop criterion, and by including more attributes or excluding some of them in the attributes subsets would yield to worse classification results. The Weka parameter settings of the implemented FS method in this work were the following: attribute evaluator CfsSubsetEval (locallyPredictive = True; missingSeparate = False; preComputeCorrelationMatrix = False) and search method best first (lookupCacheSize = 1; direction = Forward; searchTermination = 5).

We recall that the attributes Flow ID, Source and Destination IP, Source and Destination Port, and Timestamp were previously removed in the cleaning stage. The attributes selected for each attack file differentiating the specific types of attack (multiclass classification) are shown in Table 5, reducing the number of attributes considered from 77 to a range from 4–8 attributes. The selected attributes provide information of the most relevant characteristics found on every specific type of attack. In relation to DDoS-type attacks, which are characterized by suspicious amounts of traffic, attributes such as the length of the packets from source to destination or their maximum size have been selected. Moreover, the URG flag is not widely used in modern protocols and it raises certain suspicions when used (in port scanning attacks, e.g., TCP Xmas Scan, an initial packet with the PSH and URG flags is sent). Regarding the Ares Botnet attack (on Tuesday), this causes after finding a vulnerable device, Ares operators to download a version of the Ares malware on the exposed device, which then acts as another scanning point. Therefore, the number of bytes from source to destination may be significant. Within the Tuesday file for the Patator SSH attack, a successful login can be detected based on the amount of data returned by the server. This may be one reason why the number of bytes sent in one direction is relevant in this attack. The size of the window may also be related to the progressive increase in the size of password attempts.

The attributes selected when considering the complete dataset (binary classification) are shown in Table 6, in this case reducing the number of attributes from 77 to 6. Using the joint file, some of the selected attributes match those selected in specific attack files, such as Init_Win_bytes_forward or Init_Win_bytes_backward. Others like Bwd Packet Length Mean or Bwd Packet Length Std are less frequently present but also recur. If there is a significant number of attacks represented by an attribute in the complete dataset, it means that this attribute will allow the detection of a significant number of attacks more easily (Bwd Packet Length Mean and Bwd Packet Length Std). Lastly, other attributes, such as Active Mean, not included in any specific attack files can be relevant to distinguish between attack or benign traffic in a general way.

### 3.3. Classification Using Zeek for Traffic Flows Detection from the Packets in CICIDS2017 Dataset

This subsection presents the procedure followed to obtain the attributes and traffic flows using Zeek [8] applied to the raw packet captures (.pcap files) collected during CICIDS2017 dataset creation as an alternative to the use of the attributes and traffic flows available in the CICIDS2017 dataset (.csv files). Therefore, a widely used open-source data transaction logging tool is proposed instead of CICFlowMeter [27] to extract the traffic flows from the raw packet traffic captures. Recently, some authors have indicated that the CICFlowMeter tool may present some incorrect implementation aspects both in the construction of the TCP protocol flows and in the extraction of attributes [48,49], so this work provides an alternative to the study of the CICIDS2017 dataset, taking as a starting point the packet captures instead of the flows.

Zeek [8] is a well-known software, an open-source passive network traffic analyzer, which is used as a network security monitor to support investigations of suspicious activity and provides with a rich set of data logs. Hence, the objective is two-fold: firstly, to detect and label the traffic flows applying Zeek to the files that contain the direct captures of the packets generated during the construction of the CICIDS2017 dataset (traffic packet capture files); secondly, to find attributes or characteristics of such traffic flows to be used as input for the classifiers. Zeek was used to obtain the different logs from the .pcap packet capture file, executing a zeek script that allows obtaining the data of each complete connection or traffic flow (end_conn.zeek). In a real-time environment, Zeek would work directly analyzing the traffic on the network. Then, these logs are converted into .json format. The Zeek end_conn.log file was used, which contains details of each connection detected at the level of the IP, TCP, UDP, ICMP protocols (see Table 7). The .log file indicates for each flow source and destination IP addresses, source and destination ports, protocol, service, bytes sent and received, among other parameters. A conversion to .csv format is then carried out using another script developed in Python, so all that remains is to label the traffic flows as ATTACK or BENIGN, in addition to adapting the .csv file to Weka. This process of Zeek-based flow detection was applied to the 11,522.402 packets included in the CICIDS2017 dataset, yielding a file containing 445,909 traffic flows. The time required to generate the flows was approximately 4 min.

In order to label the traffic flows detected using Zeek as ATTACK or BENIGN flows, a script verifies during the time intervals of the registered attacks the source and destination IPs, source and destination ports and protocol of the detected flows. For each CICIDS2017 file, the IP addresses of the attacking computers and the time intervals in which the attacks occurred are available from the CIC [25]. For the labeling applied using Zeek, we have also taken into account some corrections for the attack time intervals suggested by [48], since there were some little discrepancies in the document provided by the CIC. As also noted in [48,49], a small number of traffic flows did not appear to be well labeled (PortScan and Bot attacks), as well as a possible bug in the DoS Hulk attack implementation. The infiltration attack was also removed because there was not enough information about its implementation to carry out the labelling process. The number of flows present in the CICIDS2017 dataset and flows detected using Zeek are shown in Table 8, using the original information from the CIC (original) and including the effect of the corrections for labelling DoS Hulk, PortScan, and Bot attacks. CICIDS2017 flows were built from the raw .pcap files using the CICFlowMeter tool [27], which defines a flow as a bidirectional exchange of network packets belonging to the same tuple of source and destination IP addresses, source and destination ports, and transport layer protocol, within a given period of time (a flow ends when it times out or when the connection is closed). Traffic flow identification using Zeek also follows the 5-tuple approach. For a connection-oriented protocol such as TCP, the definition of a connection is clearer. However, for others, e.g., UDP and ICMP, Zeek implements a flow-like abstraction to aggregate packets where each packet belongs to one connection. As a general result, the number of flows detected using Zeek is lower than the one registered in the original dataset, probably due to the time interval that CICFlowMeter considers for several packets to belong to the same connection (timeout set to 2 min) and to the effect of TCP appendices of connections, which constitute the end of the underlying TCP connection (subsequent packets belonging to the same TCP connection that constitute their own extra flow) [48].

## 4. Evaluations and Results

In this section, a concise description of the varied experimental results obtained is provided together with their interpretation. Three main test benches corresponding to the different approaches described in the previous section have been considered: First, the ML techniques are tested over the original CICIDS2017 dataset (attributes and flows as included in the .csv files), performing multiclass classification, binary classification, and classification over the complete dataset. Secondly, the ML techniques are applied to the same CICIDS2017 flows dataset but with prior selection of the most relevant attributes, performing multiclass classification and classification over the complete dataset. Finally, the ML techniques have been tested over the Zeek-derived flows and attributes (after detecting the traffic flows in the .pcap files), performing a binary classification and classification over the complete dataset.

### 4.1. Results of Classification Using the Traffic Flows in CICIDS2017 Dataset

The performance results obtained by applying ML techniques for the multiclass classification of traffic flows, i.e., differentiating between the different specific attacks recorded in the same dataset file and using the available attributes (77 features for each flow) are shown in Table 9. For each dataset file including different types of flows (benign and attack), the F1 score results for the ten classifiers are presented. As a general comment on the F1 score results, it can be said that the ML methods have reached high classification rates in most types of attacks (F1 close to 1). A better behavior is observed in the classification of BENIGN flows (with values of F1 very close to 1) with respect to the different attacks, which seems reasonable considering that the dataset files contain imbalanced classes (the number of instances of BENIGN flows with respect to attack flows is much higher, see Table 2). This is evident to a greater extent in those attacks that are less representative in the dataset, such as Heartbleed or Web Attack, where the results obtained are worse. Those results in the table with a question mark are equivalent to a value of F = 0. Weka calculates this parameter using precision and recall, and precision results in indeterminate if TP and FP are equal to 0. When comparing the results of the different techniques (Table 9) it can be seen that the performance of PART and random forest algorithms stands out, obtaining F1 values above 98% in 9 of the 14 types of attack (if the BENIGN class in each file is also considered, then F1 > 0.98 in 16 of 21 classes), and J48 algorithm in 8 of the 14 attacks (F1 > 0.98 in 15 of 21 classes, when including the BENIGN class). On the other hand, the algorithms that provided the worst classification results are naïve Bayes, MLP, SMO, and AB, with F1 values not suitable. In general, the results obtained and presented in the following tables for F1 were highly correlated with those found for the rest of performance indicators (accuracy, Cohen’s kappa coefficient, MCC, etc.). The same trend is observed in all the ML techniques for accuracy, k, or MCC as for F1 score.

An example of the analysis of the computational complexity of the ML techniques is shown in Table 10. It presents the execution times for the Friday-Afternoon DDoS file (with more than 200,000 traffic flows, distributed 50/50% in training and test subsets) both in the model construction phase (training) and the evaluation phase (test). The test execution times are also expressed in terms of number of flow/s (last row in Table 10) that the ML classifier was able to process (in a real-time operation, there would be many other factors to consider and, e.g., it would be necessary to extract the flows from the traffic and derive their attributes at least at the same rate). Those techniques with higher computational complexity led to longer execution times as expected. Although most techniques could be used in real-time in terms of ML execution times, the computational cost of some of them was very high for efficient use: logistic, SMO, IBk, and RF techniques required long times. Anyway, the absolute values presented in Table 10 should be interpreted with care since they may be highly dependent on the software implementations.

We again recall that Flow ID, Source and Destination IP addresses and ports, and Timestamp were previously removed to avoid including attributes that had been used in the flow identification process. Regarding the Destination Port attribute, which might be more controversial than the others, we decided to perform some checks regarding its role in the CICIDS2017 dataset before its removal. If the Destination Port attribute is included within the attribute set, it becomes the first ranked attribute according to its information gain for various attack files. As an example, we applied the simple rule technique (ONER) on Tuesday’s file (which contains SSH and FTP Patator attacks) including the Destination Port attribute within the attribute set and the ML technique automatically selected one rule based on this attribute that reached a classification of 99.28% of accuracy and an averaged F1 of 0.993. These performance results appear to be too optimistic unless the algorithm has learnt the ports used in the attacks during the training phase, as it seems the case for the CICIDS2017 dataset. Other tests carried out with other ML techniques also corroborated the increase in the performance obtained when including the Destination Port. Therefore, despite the fact that the results obtained might be slightly worse, we decided to also remove the Destination Port attribute from the set of attributes in all the experiments carried out in this work in order to build the classification models as general as possible.

After realizing that some classification errors found corresponded to the incorrect classification of a specific type of attack as another attack (in those files that contain more than one type of attack), an alternative way for evaluating the ML techniques was considered. The following example illustrates how considering binary classification (distinguishing only between normal traffic and attack) versus multiclass classification (distinguishing between different types of attack) permits to improve the results in some cases. The confusion matrix in Table 11 shows a significant example that corresponds to the application of the J48 technique on the Thursday Morning Web Attacks file, where most of the errors detected were due to multiclass classification and not to incorrect classification of benign flows as attacks and vice versa. In the row of a = BENIGN (first row) all the flows that are outside the first cell correspond to false alarms (FP), classifying benign traffic as some type of attack (20 flows of 84,066). On the other hand, in the first column (a), all the flows outside the first cell (22 flows) represent attacks classified as benign (FN). All flows outside the main diagonal in the confusion matrix are counted as misclassified flows, but those errors that are neither in the first row nor in the first column of the matrix correspond to errors in the classification of the specific attack type. In Table 11 a total of 332 flows were misclassified but 330 XSS attacks were classified as brute force attacks, which should perhaps be considered a less critical error than when a FP or FN occurs. This reasoning led us to consider the performance obtained in binary classification i.e., when the classifier makes a binary distinction between BENIGN and ATTACK flows. The confusion matrix obtained in this case using the J48 algorithm on the Thursday Morning Web Attacks file is shown in Table 12, where the number of instances labeled as ATTACK that have been classified as such has increased considerably with respect to the results in Table 11. The classification errors are now significantly reduced from 374 to only 38 (from 0.439% to 0.044%).

The F1 performance results obtained in binary classification are shown in Table 13. In those files containing more than one attack type, all attacks were combined in the ATTACK class (in the rest of the files, obviously, no changes will be produced, but the results are included to facilitate the comparison). From a general point of view, an improvement in the detection of the ATTACK class is observed, due to many classification errors in multiclass classification occurred among the different types of attack. This improvement is more significant in MLP, AB, and OneR algorithms, since they were not able to distinguish between certain attacks, and by grouping them into one class improved their performance. Using the PART algorithm in the binary classification, values of F1 > 0.98 were obtained (see Table 13) in 13 out of 14 total classes (including ATTACK and BENIGN classes present in each of the seven files), and only the Infiltration attack presented a lower value (F1 = 0.762). In the multiclass classification, this performance (F1 > 0.98) was achieved in 16 of 21 total classes at maximum. After PART algorithm, the best results were obtained using the J48 and random forest algorithms, showing again that tree-based techniques achieve the best results (in this case, RF improves less than the other two algorithms). Moreover, the execution time has also been reduced, since the complexity of the classification is reduced to differentiating between only BENIGN and ATTACK classes. Algorithms, such as SMO or IBk, continue to present high execution times, with logistic and naive Bayes being in the test stage the ones that have significantly reduced their time.

Finally, all the flows present on the different files were grouped into a complete dataset file (all files) and a binary classification was performed. The joint file obtained from the CICIDS2017 dataset includes 2830.743 traffic flows (2273.097 of BENIGN class and 557.646 of ATTACK class). This grouping includes all the files: Tuesday, Wednesday, Thursday, Friday, and Monday (which only contains benign traffic). In this way, it is possible to study how the algorithms behave in the face of a combination of attacks with different characteristics. Under this approach, a unique classification model is produced for the complete dataset instead of one model for each file (and its corresponding attacks). The F1 performance results obtained in the complete dataset are shown in Table 14. The algorithms that presented the best F1 values were again PART, J48, and random forest, presenting values of F1 very close to 1. In this case, there is a slight decrease (from 1 to 0.999) in the value of F1 in the detection of the BENIGN class, which may be due to the fact that the algorithms, faced with a greater variety of attacks, build a more general model, and therefore this increases the probability of considering benign traffic as an attack. Regarding other performance scores, the highest MCC values were reached by PART, J48, and random forest (0.997, 0.997, and 0.996, respectively), which shows that the classifiers present a very symmetrical behavior with respect to the traffic classes involved.

### 4.2. Results of Classification Using the Traffic Flows in CICIDS2017 Dataset and Attributes Selection

The classification results obtained using the ML techniques with prior attributes selection following the methodology described in Section 3.2 are presented and analyzed now. In multiclass classification, the attributes were selected individually for each dataset file (attack) as shown in Table 5. The F1 performance results with attributes selection are presented in Table 15, where it is possible to see that the best performance was achieved again by PART, J48, and random forest algorithms, all with 13 out of 21 classes with F1 > 0.98. In all the files, the ML techniques reached F1 values above 0.999 for the BENIGN class, and high values for the ATTACK class in those files with only one type of attack (binary classification), e.g., DDoS and PortScan present in the Friday Afternoon files. However, algorithms, such as Logistic, SMO, or AdaBoost, have shown a drastic F1 score decrease on certain files (see Table 15), specifically, in Friday Morning Bot, Thursday Morning Web Attacks, Tuesday, and Wednesday DoS (files with more than two classes). A reduction in the number of attributes may have eliminated some attribute relevant to a specific attack with few instances. Again, the same trends of results were observed for accuracy, k, or MCC as for F1.

On the other hand, the computation time was also noticeably reduced when the number of attributes was lower. As an example, Table 16 shows the decrease of the execution times (in percentage) obtained with attribute selection for the Friday-Working Hours-Afternoon DDoS file. Those techniques with greater computational complexity gave rise to higher execution times, but the times were reduced by using a smaller number of attributes, in a significant manner for most algorithms.

Now, the binary classification analysis based on the grouping of all the traffic flow files in the CICIDS2017 dataset with the previous selection of the most significant attributes is presented. This allows to study how the F1 scores vary with respect to the tests carried out in Section 4.1 using all attributes. Table 17 shows the F1 performance results obtained for the different ML techniques, and again the tree-based algorithms (PART, J48, and random forest) that have stood out in previous tests are the ones that obtained the best results. It can be seen that all techniques except logistic and AdaBoost maintain very similar F1 scores with respect to those obtained using all the attributes (see Table 14), making it possible to reduce the execution time (drastically in most of the algorithms) by selecting the most relevant ones, maintaining a high classification capacity. Regarding other performance scores, the highest MCC values were again reached for PART, J48, and random forest with an identical value of 0.970, hence showing that the classifiers in this configuration continue to exhibit a symmetric behavior with respect to traffic classes.

### 4.3. Results of Classification Using Zeek for Flows Detection from Packets in CICIDS2017 Dataset

This subsection presents the classification results obtained using the flows and attributes estimated after the application of Zeek as described in 3.3. Hence, the performance of a conventional tool like Zeek can be studied in order to use it in further real-time implementations, where Zeek would analyze the traffic in the network instead of the stored .pcap capture files. The attributes extracted using Zeek shown in Table 7 are considered (except source and destination IP addresses, source and destination ports, flow ID and TimeStamp, since all of them have been used to define a traffic flow or are associated with packets belonging to it). The classification has been carried out by defining the training and test subsets as in previous subsections using 50/50% for training and testing. The F1 score results obtained by applying the ML techniques using Weka over the Zeek-based flows are shown in Table 18, where each file corresponds to a single day and includes different types of attacks (as collected in the .pcap files in the CICIDS2017 dataset) labelled as a generic ATTACK class. The results presented in Table 18 correspond to the tests carried out considering all the corrections for the traffic flows labelling described in Section 3.3: in Wednesday file, the DoS Hulk is not included; the Thursday file does not include the infiltration attack; and certain PortScan and Bot flows that were not well tagged have been eliminated in the Friday file. The performance results obtained using Zeek-based flows and attributes were very high with F1 values close to one, mainly for tree-based techniques (PART and J48). The results obtained stand out in the attacks corresponding to Friday file (DDoS, Bot and PortScan attacks) and Tuesday file (FTP-Patator and SSH-Patator attacks) while the classification is slightly worse in Thursday file (Web attacks). Regarding the execution times, the fastest algorithm was J48 (for instance, it required 12.79 s to build the model and 0.2 s to perform the test in the Friday file), while the times required to build the model and perform the test were, respectively, 22.07 and 0.39 s for PART, and 138.21 and 4.68 s for RF, respectively.

Finally, all the flows detected using Zeek in the different files were grouped into a complete dataset file in a similar way than in previous subsections and a binary classification was performed. The joint file obtained using Zeek includes 1323.590 traffic flows (1030.313 of BENIGN class and 293.277 of ATTACK class). The F1 results are presented in Table 19 where it is shown that again the tree-based ML techniques achieved the highest classification rates, although it is notable that some other algorithms (e.g., IBk, SMO and Logistic) improved their performance compared to the previous binary classifications in the entire dataset (see Table 14 and Table 17). As in previous Section 4.1 and Section 4.2, a single classification model is generated for the entire dataset and the techniques have to deal with a combination of attacks with different characteristics. Despite this, the results continued to reach very high values of F1, showing that Zeek-based attributes can be very appropriate to characterize the traffic flows and they allowed to classify even better than using the original attributes included in the CICIDS2017 .csv files (see Table 17). As in previous tests, the same result trends were observed for accuracy, k, or MCC as for F1. The highest MCC values were reached for PART (0.991), J48 (0.991), and RF (0.989).

Finally, in order to evaluate the effect of the corrections in the traffic flows labelling described in Section 3.3, a comparison between the results obtained for the tree-based techniques using the different labelling versions for the Friday and Wednesday files is shown in Table 20 and Table 21. As can be seen, the differences in F1 values considering the different flow labelling corrections are very small for the Friday file, since the number of flows that were corrected is not large, although some classification improvement was obtained using the proposed labelling refinement. However, for the Wednesday file, the results got worse when the Hulk DoS attacks were not included in the dataset. As stated in [48], the implementation of the Hulk DoS attack might not be correct, but we can infer from these results that despite this, the structure of the Hulk DoS flows in the dataset may be similar to that of other attack flows considered and their inclusion may help to reinforce a better classification.

## 5. Discussion

### 5.1. General Observations about the Results

In this work, different machine learning techniques (naive Bayes, logistic, multilayer perceptron, SMO, K-NN, AdaBoost, OneR, PART, J48, and random forest) have been applied to model the traffic connections and distinguish between those corresponding to attacks and normal flows in the CICIDS2017 dataset. Several approaches have been considered to evaluate the classification methods: First, the ML techniques were tested over the original CICIDS2017 dataset (attributes and flows as included in the .csv files) performing multiclass classification, binary classification, and binary classification over the complete dataset. Secondly, the ML techniques were applied to the same CICIDS2017 flows dataset but with prior selection of the most relevant attributes, performing multiclass classification and binary classification over the complete dataset. Finally, the ML techniques were tested over the Zeek-derived flows and attributes (after detecting the traffic flows in the .pcap files), performing a binary classification and binary classification over the complete dataset.

From the results obtained after carrying out different classifications with and without attribute selection, it can be stated that the most suitable classifiers (obtaining the best F1 results and with execution times allowing several hundred thousand flows/s to be processed) were those based on decision trees: PART, J48, and random forest (RF requiring 20 times longer execution times than PART or J48). Tree-based technique implementations are the most easily interpretable since they behave like rule-based systems. These algorithms obtained F1 values greater than 0.999 (average values in the tests carried out in binary classification on the complete dataset file) without attribute selection (77 attributes were considered). The results obtained in binary classification have shown that by reducing the classification complexity, better results can be obtained in less time. In the approach of multiclass classification, it was observed that some of the classification errors found corresponded to the incorrect classification of a specific type of attack as another attack (e.g., between XSS and Brute Force web attacks). Therefore, a simplification could be assumed through binary classification in which all attacks were grouped into a single class to facilitate the task of classifiers. It is noteworthy that the use of a single classification model generated for the entire data set (including all attacks) instead of specific models for each attack allowed reaching very high F1 values. The results corroborate the generally accepted statement that the classification rate for a two-class or binary IDS is higher than that of multi-attack IDS due to the number of classes the ML technique has to learn to make correct predictions, and due to the nature of data, wherein imbalance can be more evident in multi-attack datasets than in two-class datasets [40].

Feature selection could be essential in a real-time operation since a reduced time to generate or update the model and mainly to perform the test would be needed. The reduction in execution time was notable (above 30% and up to 58% of decrease in the time required to test the model for the tree-based techniques) at the expense of a slight decrease in the value of F1, reaching values above 0.990 with CFS-based attribute selection (six attributes selected) for the tree-based algorithms. Unlike other research works, our proposal is based on an attribute selection method (CFS in combination with best first) that considers not the discriminatory power of the attributes ranking them individually by measuring e.g., the information gain with respect to the class, but instead seeks to find the best subset of attributes to work together (taking into account their correlation or redundancy). An important aspect regarding feature selection when comparing different works is whether the attributes used to extract the traffic flows during the dataset creation (5-tuple features) have been also considered to build the classification model or not. Attributes from the 5-tuple, e.g., source and destination IP addresses and ports, can be very dependent on the dataset and might not be relevant in a generic model. The inclusion of such parameters can facilitate the classification of traffic flows and therefore increase the performance of ML techniques, but probably at the cost of generating overfitting classifiers.

Finally, as a novelty, in this work, we have proposed to use a common tool, Zeek, to extract the traffic flows contained in the raw traffic packet captures and to derive new attributes from these flows. By means of Zeek, the flows were identified, and the network connections information was collected in the end_conn.log file. The best classification results were obtained again using the PART, J48, and random forest algorithms, with high values of F1 (above 0.980 in most types of attack, yielding an averaged F1 of 0.997 in the entire dataset). The execution times were also low (allowing several hundred thousand flows/s to be processed) thanks to the use of only 14 attributes.

### 5.2. Comparison with Related Work

As has already been said, many research works have been proposed in the application of ML techniques for intrusion detection using different datasets. Some reviews of those works can be found in [4,9,10,11,12,13]. On the other hand, some recently published studies [25,48,49,50,51,52] have applied ML techniques over the CICIDS2017 dataset (most of them using the available attributes and flows as included in the .csv files). The comparison of the results obtained in this work and others is summarized in Table 22 and below some explanations about them are provided. The comparison table shows for each referenced work the best F1 score results reached using a determined ML technique, and the FS method used and number of attributes selected (N attr.) if this information is provided in the paper.

In addition to the description of the methodology followed in the creation of the CICIDS2017 dataset, some ML algorithms were evaluated in [25] for the detection of the different attack categories. A feature selection for each attack was carried out using the random forest algorithm, and then the following techniques were applied: K-nearest neighbors (KNN), random forest (RF), ID3 (precursor algorithm of C4.5 used in this work), Adaboost (AB), multilayer perceptron (MLP), naive-Bayes (NB), and quadratic discriminant analysis (QDA), with F1 results of 0.96/0.97/0.98/0.77/0.76/0.84/0.92, respectively. The results obtained in our work (with F1 above 0.99) are not straight-forward comparable because not all the information on the evaluation process followed is available in [25] where a weighted average of evaluation metrics was applied. However, a correlation of those results is observed in that the techniques with the best performance correspond to ID3 and RF (tree-based techniques), and the worst to MLP, AB, and naive Bayes.

The work in [50] used the CICIDS2017 dataset to compare the efficiency of two dimensionality reduction approaches with different classification algorithms, such as random forest, Bayesian network, LDA, and QDA in binary and multi-class classification. That technique reduced the CICIDS2017 features from 81 to 10, while maintaining a high accuracy in multi-class and binary classification. However, although the work is interesting, those results were obtained including within the selected attributes the Source and Destination IP addresses as well as the Source and Destination Ports (only FlowID and Timestamp were removed), which as previously discussed form the 5-tuple that defines a flow.

In [51], the information gain was used to select the most significant features, and various ML techniques were subsequently implemented. The random forest algorithm had the highest accuracy of 99.86% using 22 selected features and 99.81% using 15 features, whereas the J48 classifier provided an accuracy of 99.87% using 52 features with longer execution time. In that work, the experiments carried out were performed with only 20% of the flows in the CICIDS2017 dataset, thus reducing the number of instances (in our work, the full dataset was considered) and under different validation conditions (70/30% split for training and test subsets vs. 50/50% split in our work). The destination port attribute was included, which was selected as the fourth attribute in the information gain feature ranking selection and in all the attribute sets used in that work. Regarding the effect of feature selection, results in [51] corroborate that by reducing the number of attributes the accuracy decreases but the execution times are also reduced.

The work published in [52] analyzed 30 percent of the CICIDS2017 dataset. Different attributes were selected by means of an ensemble classifier to subsequently feed different classifiers. Among the selected characteristics, both the destination port and the source port were considered, which turned out to be the most relevant attributes facilitating the identification of flow attacks in the dataset. The classifiers with the best results were decision trees (random forest and bagging classifiers), reaching F1 values of 0.999 in the reduced dataset.

The interesting work presented in [48] revisits the CICIDS2017 dataset and its data collection and analyses, correctness, validity, and overall utility of the dataset for the learning task. They regenerated and relabeled traffic flows to correct possible errors in the CICFlowMeter tool and evaluated a random forest (RF) classifier on the original and corrected datasets. The train-test split used was 75/25%, respectively, and the F1 results obtained ranged from 0.79 up to 1 depending on the attack type (weighted average of F1 = 0.99). In that work, the number of flows considered was lower than that present in the original CICIDS2017 dataset (generated with CICFlowMeter), in line with the results we have obtained when extracting the flows using Zeek.

Finally, in [49], a similar work analyzed this dataset in detail and reported several problems with the flows included in the CICIDS2017 .csv files (generated with CICFlowMeter). To address these issues, they proposed a flow extraction tool to be used in the CIC-IDS2017 raw data files. Among others, the destination port attribute was included in their study. Several machine learning algorithms were tested, obtaining the best results with decision tree and random forest classifiers (F1 > 0.998).

As a general summary of this comparison, it can be said that the work presented here has in common with the previous ones the high classification performances obtained by algorithms based on trees. In our work, however, algorithms, such as PART or J48, may offer a faster alternative solution to the RF technique (mainly used in related works). Moreover, in our proposal, we have not included any of the 5-tuple attributes which define a flow in the classification models, while some of them have been included in other works.

### 5.3. Threats to the Validity

Regarding the threats to the validity of our study, following the principles described in [44,45], the following issues can be identified:External validity: It addresses the generalizability of the research experimental process. The quality of the results obtained depends on the dataset used: the CICIDS2017 dataset. It was selected because it includes a wide range of up-to-date attacks and because it was developed considering most of the characteristics an IDS dataset should include. However, the experimental process should be rerun over different datasets to guarantee the generalizability of the classification models since results in this kind of studies can be very good (F1 values close to 1) but highly dependent on the dataset. On the other hand, the techniques used in this work for traffic flow-based intrusion detection could be generalizable to other subjects that share common properties, to name a few, in the area of software defect prediction models to predict the quality and reliability of a software system, or in data leakage protection systems (DLPs) to classify information in confidential or non-confidential categories, or to identify altered documents, since the nature of the inherent problem is associated with classification and feature selection tasks.Internal validity: It refers to the choice of prediction models and feature selection methods. In this work, ten classifiers and many different approaches were considered in the benchmark to make the study as broad as possible, including the use of the original labelled traffic flows in the dataset and the Zeek-based flows from the raw packet captures, the evaluation considering multi-class vs. binary classifications, the application of FS methods, etc. Anyway, some changes in attribute selection or tree models tuning will probably be needed if the models are applied to other datasets and more sophisticated classifiers and FS methods could also be deployed.Construct validity: It focuses on the choice of indicators used to evaluate the performance of classification models. Regarding this aspect, we estimated the following parameters for every classification test in the benchmark study: F1 score, accuracy, precision, recall, Cohen’s kappa coefficient, and MCC, among others. For the sake of simplicity and conciseness, we have selected the F1 score results instead of others because it represents a combined value of precision and recall, and thus it is the most commonly used in the literature to present traffic classification results in IDS. However, we suggest that the use of other indicators, such as Cohen’s kappa coefficient or MCC, could be more appropriate to estimate the classifier performance (the improvement provided with respect to the accuracy that would occur by mere chance) and with a symmetric behavior in imbalanced classes.

## 6. Conclusions and Future Work

The results obtained in this work allow us to affirm that tree-based machine learning techniques may be appropriate in the flow-based intrusion detection problem, having shown better classification rates than other more complex algorithms and having required lower execution times (allowing several hundred thousand flows/s to be processed). The classification scores obtained in the CICIDS2017 dataset showed F1 values above 0.999 (average values in the tests performed in binary classification on the entire dataset using 77 attributes), F1 above 0.990 with CFS-based attribute selection (average values in binary classification on the entire dataset with six selected attributes), and F1 above 0.997 using Zeek-derived flows and attributes (average values in binary classification across the entire dataset using 14 attributes). Among the tree-based techniques, algorithms, such as PART or J48, can offer a faster alternative solution to random forest.

As future research, the analysis of other Zeek logs will be considered to extract more features to be included in the classification models. On the other hand, in order to validate the generalizability of the proposed classification models, they should be tested on other different IDS datasets since, although the results obtained have been very good (F1 values close to 1), they may be dependent on the dataset.

## Figures and Tables

**Figure 1 sensors-22-09326-f001:**
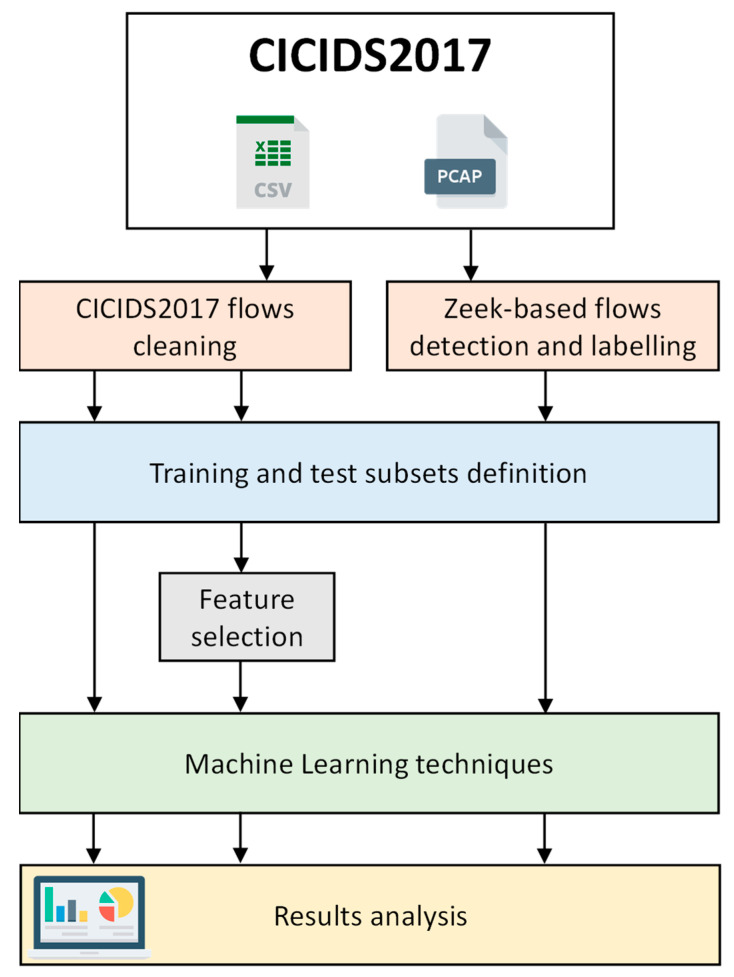
General methodology workflow.

**Table 1 sensors-22-09326-t001:** Description of files and flows included in the CICIDS2017 dataset.

Files	Benign Flows and Attacks
Monday-Working Hours	Benign (Normal human activities)
Tuesday-Working Hours	Benign, FTP-Patator, SSH-Patator
Wednesday-Working Hours	Benign, DoS GoldenEye, DoS Hulk, DoS Slowhttptest, DoS slowloris, Heartbleed
Thursday-Working Hours-Morning-WebAttacks	Benign, Web Attack-Brute Force, Web Attack-SQL Injection, Web Attack-XSS
Thursday-Working Hours-Afternoon-Infilteration	Benign, Infiltration
Friday-Working Hours-Morning	Benign, Bot
Friday-Working Hours-Afternoon-PortScan	Benign, PortScan
Friday-Working Hours-Afternoon-DDoS	Benign, DDoS

**Table 2 sensors-22-09326-t002:** Instance occurrence for the different flows in the CICIDS2017 dataset.

Labeled Flow	Number of Flows
Benign	2,359,087
DoS Hulk	231,072
PortScan	158,930
DDoS	41,835
DoS GoldenEye	10,293
FTP-Patator	7938
SSH-Patator	5897
DoS Slowloris	5796
DoS Slowhttptest	5499
Bot	1966
Web Attack-Brute Force	1507
Web Attack-XSS	652
Infiltration	36
Web Attack-SQL Injection	21
Heartbleed	11

**Table 3 sensors-22-09326-t003:** Computational complexity for different ML techniques.

Algorithm		Train Time Complexity	Test Time Complexity
Naive Bayes	c number of classes	O(n⋅m⋅c)	O(m⋅c)
Logistic regression model (Logistic)		O(n⋅m^2^ + m^3^)	O(m)
Multi-layer perceptron (MLP)	c number cycles, k number neurons	O(c⋅n⋅m⋅k)	O(m⋅k)
Support vector classifier (SMO)	s number of support vectors	O(m⋅n^2^ + n^3^)	O(s⋅m)
K-nearest neighbours classifier (IBk)	k number of neighbours	O(1)	O(n⋅m)
Adaboost classifier (AB)	e number of estimators	O(n⋅m⋅e)	O(m⋅e)
1R classifier (ONER)		<O(n⋅m⋅log n)	O(1)
Partial C4.5 decision tree (PART)	k depth of tree	O(n⋅m⋅log n)	O(k)
C4.5 decision tree (J48)	k depth of tree	O(n^2^⋅m)	O(k)
Random Forest (RF)	t number of trees k depth of tree	O(n⋅m⋅t⋅log n)	O(k⋅t)

**Table 4 sensors-22-09326-t004:** Confusion matrix for anomaly traffic classification.

Class\Prediction	Attack	Normal
Attack	TP	FN
Normal	FP	TN

**Table 5 sensors-22-09326-t005:** Attributes selected considering the different types of attack (multiclass classification).

File Name	Attributes
Friday Afternoon DDoS	Total Length of Fwd Packets, Total Length of Bwd Packets, Fwd Packet Length Max, Bwd Packet Length Min, URG Flag Count, Subflow Bwd Bytes, Init_Win_Bytes_Forward, Min_Seg_Size_Forward
Friday Afternoon PortScan	Bwd Packet Length Mean, PSH Flag Count, Init_Win_Bytes_Backward, Act_Data_Pkt_Fwd, Min_Seg_Size_Forward
Friday Morning Bot	Bwd Packet Length Mean, Bwd Packet Length Std, Fwd IAT Max, Packet Length Std, Avg Bwd Segment Size, Init_Win_Bytes_Backward, Min_Seg_Size_Forward
Thursday Afternoon Infiltration	Total Length of Fwd Packets, Active Std, Active Min, Idle Std
Thursday Morning Web Attacks	Fwd Packet Length Mean, Fwd IAT Min, Init_Win_Bytes_Backward
Tuesday	Fwd Packet Length Mean, Fwd Packet Length Std, Bwd Packet Length Std, Fwd PSH Flags, Avg Fwd Segment Size, Init_Win_Bytes_Forward, Init_Win_Bytes_Backward, Min_Seg_Size_Forward
Wednesday DoS	Total Length of Bwd Packets, Bwd Packet Length Mean, Min Packet Length, Init_Win_Bytes_Forward, Init_Win_Bytes_Backward, Idle Max

**Table 6 sensors-22-09326-t006:** Attributes selected in the complete dataset (binary classification).

File Name	Attributes
All files	Bwd Packet Length Mean, Bwd Packet Length Std, Packet Length Std, Init_Win_Bytes_Forward, Init_Win_Bytes_Backward, Active Mean

**Table 7 sensors-22-09326-t007:** Attributes collected using Zeek (end_conn.log file).

Attributes		
TimeStamp	Duration	Source IP bytes
Source IP	Source Bytes	Response Packets
Source Port	Response Bytes	Response IP Bytes
Destination IP	Conn_state	Tunnel parents
Destination Port	Missed Bytes	Label
Protocol	History	
Service	Source Packets	

**Table 8 sensors-22-09326-t008:** Number of flows in the CICIDS2017 dataset and extracted using Zeek.

File	CICIDS2017 Flows	Zeek Flows
Tuesday	445,909	322,676 (original)
Wednesday	691,703	508,801 (original with DoS Hulk) 345,347 (without DoS Hulk)
Thursday	170,366	113,005 (without Infiltration)
Friday	704,245	546,795 (original) 542,567 (PortScan fixed) 546,790 (Bot fixed) 542,562 (PortScan and Bot fixed)

**Table 9 sensors-22-09326-t009:** F1 performance results obtained for each dataset file in multiclass classification.

File	Label	NB	Logist	MLP	SMO	IBk	AB	ONER	PART	J48	RF
Friday Afternoon DDoS	BENIGN	0.937	0.999	0.980	0.980	1	0.997	0.988	1	1	1
	DDos	0.956	0.999	0.986	0.985	1	0.998	0.991	1	1	1
Friday Afternoon PortScan	BENIGN	0.898	0.999	0.992	0.995	1	0.998	0.995	1	1	1
	PortScan	0.929	0.999	0.994	0.996	1	0.998	0.996	1	1	1
Friday Morning Bot	BENIGN	0.904	0.995	0.995	0.996	0.999	0.998	0.998	1	1	1
	Bot	0.107	0.995	0.029	0.397	0.938	0.709	0.727	0.981	0.966	0.971
Thursday Afternoon Infiltration	BENIGN	0.997	1	1	1	1	1	1	1	1	1
	Infiltration	0.025	0.389	? = 0	0.353	0.500	? = 0	0.211	0.762	0.444	0.667
Thursday Morning Web Attacks	BENIGN	0.925	0.999	0.993	0.993	1	0.993	0.998	1	1	1
	Web Attack Brute Force	0.021	0.769	0	0	0.728	? = 0	0.685	0.811	0.812	0.750
	Web Attack XSS	0.417	0.102	? = 0	? = 0	0.385	? = 0	0.038	0.104	0.102	0.331
	Web Attack SQL Injection	0.009	0.105	? = 0	? = 0	0.111	? = 0	? = 0	0.667	0.455	0.133
Tuesday	BENIGN	0.878	0.996	0.984	0.990	1	0.984	0.996	1	1	1
	FTP Patator	0.262	0.989	? = 0	0.587	0.997	? = 0	0.987	0.999	0.998	0.999
	SSH Patator	0.189	0.651	? = 0	0.633	0.986	? = 0	0.634	0.998	0.994	0.997
Wednesday DoS	BENIGN	0.755	0.989	0.96	0.965	0.999	0.96	0.954	1	0.999	0.999
	DoS slowloris	0.255	0.978	? = 0	0.875	0.989	? = 0	0.386	0.992	0.992	0.994
	DoS Slowhttptest	0.234	0.948	? = 0	0.859	0.974	? = 0	? = 0	0.981	0.982	0.985
	DoS Hulk	0.814	0.981	0.945	0.943	0.999	0.934	0.946	1	0.999	0.999
	DoS GoldenEye	0.422	0.981	? = 0	0.939	0.996	? = 0	0.400	0.996	0.995	0.996
	Heartbleed	0.833	0.480	? = 0	0.923	0.833	? = 0	? = 0	0.833	0.923	1

Note: ? = 0 means F1 equivalent to 0, for those cases where TP = FP = 0.

**Table 10 sensors-22-09326-t010:** Execution times (in training and test phases) in the Friday-Afternoon DDoS file.

	NB	Logist	MLP	SMO	IBk	AB	ONER	PART	J48	RF
Time to build the model (s)	2.52	741	103	170	0.06	62.61	3.58	76.62	48.65	171
Time to test the model (s)	3.91	0.47	0.29	0.31	1730	0.24	0.21	0.14	0.11	2.85
Number of flows/s classified in test phase	28,867	240,153	389,214	364,103	65	470,300	537,485	806,228	1,026,109	39,604

**Table 11 sensors-22-09326-t011:** Confusion Matrix for J48 on Thursday Morning Web Attacks (multiclass classification).

a	b	c	d	← Classified as
84,046	5	13	2	a = BENIGN
6	738	0	0	b = Web Attack Brute Force
8	330	20	1	c = Web Attack XSS
8	1	0	5	d = Web Attack SQL Injection

**Table 12 sensors-22-09326-t012:** Confusion Matrix for J48 on Thursday Morning Web Attacks (binary classification).

a	b	← Classified as
84,048	18	a = BENIGN
20	1097	b = ATTACK

**Table 13 sensors-22-09326-t013:** F1 performance results obtained for each dataset file in binary classification.

File	Label	NB	Logist	MLP	SMO	IBk	AB	ONER	PART	J48	RF
Friday Afternoon DDoS	BENIGN	0.937	0.999	0.980	0.980	1	0.997	0.988	1	1	1
	DDos	0.956	0.999	0.986	0.985	1	0.998	0.991	1	1	1
Friday Afternoon PortScan	BENIGN	0.898	0.999	0.992	0.995	1	0.998	0.995	1	1	1
	PortScan	0.929	0.999	0.994	0.996	1	0.998	0.996	1	1	1
Friday Morning Bot	BENIGN	0.904	0.995	0.995	0.996	0.999	0.998	0.998	1	1	1
	Bot	0.107	0.995	0.029	0.397	0.938	0.709	0.727	0.981	0.966	0.971
Thursday Afternoon Infiltration	BENIGN	0.997	1	1	1	1	1	1	1	1	1
	Infiltration	0.025	0.389	? = 0	0.353	0.500	?	0.211	0.762	0.444	0.667
Thursday Morning Web Attacks	BENIGN	0.926	0.999	0.993	0.996	1	0.996	0.998	1	1	1
	ATTACK	0.159	0.944	0	0.747	0.969	0.744	0.856	0.987	0.983	0.974
Tuesday	BENIGN	0.878	0.996	0.984	0.991	1	0.989	0.996	1	1	1
	ATTACK	0.227	0.868	? = 0	0.644	0.993	0.429	0.866	0.998	0.996	0.998
Wednesday DoS	BENIGN	0.871	0.970	0.970	0.965	0.999	0.979	0.954	1	0.999	0.999
	ATTACK	0.806	0.949	0.947	0.941	0.999	0.964	0.914	0.999	0.999	0.999

Note: ? = 0 means F1 equivalent to 0, for those cases where TP = FP = 0.

**Table 14 sensors-22-09326-t014:** F1 performance results in binary classification for the complete dataset.

File	Label	NB	Logist	MLP	SMO	IBk	AB	ONER	PART	J48	RF
All files	BENIGN	0.709	0.961	0.935	0.954	0.999	0.964	0.966	0.999	0.999	0.999
	ATTACK	0.516	0.853	0.602	0.807	0.996	0.829	0.841	0.998	0.998	0.997

**Table 15 sensors-22-09326-t015:** F1 performance results in multiclass classification using attributes selection.

File	Label	NB	Logist	MLP	SMO	IBk	AB	ONER	PART	J48	RF
Friday Afternoon DDoS	BENIGN	0.750	0.966	0.964	0.942	0.999	0.998	0.988	0.999	0.999	0.999
	DDos	0.868	0.976	0.974	0.960	0.999	0.998	0.991	0.999	0.999	1
Friday Afternoon PortScan	BENIGN	0.899	0.987	0.991	0.988	0.999	0.991	0.992	0.999	0.999	0.999
	PortScan	0.930	0.990	0.993	0.990	0.999	0.993	0.994	0.999	0.999	0.999
Friday Morning Bot	BENIGN	0.794	0.995	0.995	0.995	0.998	0.995	0.998	0.998	0.998	0.998
	Bot	0.056	0.020	0.030	? = 0	0.792	? = 0	0.727	0.793	0.794	0.795
Thursday Afternoon Infiltration	BENIGN	0.995	1	1	1	1	1	1	1	1	1
	Infiltration	0.015	0.435	? = 0	0.250	0.385	? = 0	0.303	0.500	0.3	0.667
Thursday Morning Web Attacks	BENIGN	0.229	0.993	0.993	0.993	1	0.993	0.998	1	1	1
	Web Attack Brute Force	0.001	? = 0	? = 0	? = 0	0.765	? = 0	0.685	0.801	0.803	0.758
	Web Attack XSS	0.393	? = 0	? = 0	? = 0	0.265	? = 0	0.038	0.112	0.113	0.283
	Web Attack SQL Injection	0.008	? = 0	? = 0	? = 0	0.125	? = 0	? = 0	0.400	0.222	0.133
Tuesday	BENIGN	0.970	0.984	0.984	0.984	1	0.984	0.996	1	1	1
	FTP Patator	0.945	0	? = 0	? = 0	0.999	? = 0	0.955	0.998	0.999	0.999
	SSH Patator	0.321	0	? = 0	? = 0	0.997	? = 0	0.634	0.992	0.992	0.998
Wednesday DoS	BENIGN	0.790	0.920	0.900	0.898	0.999	0.899	0.954	0.999	0.999	0.999
	DoS slowloris	0.309	0.100	? = 0	? = 0	0.956	? = 0	0.386	0.958	0.956	0.960
	DoS Slowhttptest	0.035	0	? = 0	? = 0	0.929	? = 0	? = 0	0.926	0.930	0.932
	DoS Hulk	0.787	0.882	0.782	0.779	0.999	0.782	0.946	0.999	0.999	0.999
	DoS GoldenEye	0.226	0.087	? = 0	? = 0	0.986	? = 0	0.400	0.993	0.991	0.993
	Heartbleed	0.600	1	? = 0	? = 0	0.923	? = 0	? = 0	0.600	0.600	0.600

Note: ? = 0 means F1 equivalent to 0, for those cases where TP = FP = 0.

**Table 16 sensors-22-09326-t016:** Decrease in percentage of execution times (in training and test phases) in the Friday-Afternoon DDoS file using attributes selection.

	NB	Logist	MLP	SMO	IBk	AB	ONER	PART	J48	RF
Decrease in time to build the model (%)	89	99	71	70	67	94	94	96	95	68
Decrease in time to test the model (%)	86	72	66	74	19	63	71	36	45	58

**Table 17 sensors-22-09326-t017:** F1 results in binary classification in complete dataset using attributes selection.

File	Label	NB	Logist	MLP	SMO	IBk	AB	ONER	PART	J48	RF
All files	BENIGN	0.898	0.930	0.935	0.934	0.994	0.934	0.966	0.994	0.994	0.994
	ATTACK	0.504	0.577	0.605	0.602	0.976	0.603	0.841	0.976	0.976	0.976

**Table 18 sensors-22-09326-t018:** F1 performance results using Zeek derived flows and attributes.

File	Label	NB	Logist	MLP	SMO	IBk	AB	ONER	PART	J48	RF
Tuesday	BENIGN	0.859	1	0.989	1	1	1	1	1	1	1
	ATTACK	0.152	0.998	? = 0	0.998	0.998	0.996	0.987	0.997	0.997	0.998
Wednesday (without Hulk)	BENIGN	0.891	0.999	0.958	0.998	0.999	0.994	0.996	0.999	0.998	0.999
	ATTACK	0.471	0.985	? = 0	0.974	0.987	0.931	0.961	0.986	0.980	0.986
Thursday (without Infiltration)	BENIGN	0.892	0.997	0.991	0.991	0.995	0.996	0.996	0.997	0.997	0.996
	ATTACK	0.160	0.844	? = 0	? = 0	0.760	0.790	0.834	0.872	0.876	0.801
Friday (PS and BOT fixed)	BENIGN	0.905	0.997	? = 0	0.996	0.999	0.994	0.996	0.999	0.999	0.999
	ATTACK	0.912	0.997	0.642	0.995	0.999	0.993	0.995	0.999	0.999	0.999

Note: ? = 0 means F1 equivalent to 0, for those cases where TP = FP = 0.

**Table 19 sensors-22-09326-t019:** F1 performance results using Zeek derived flows and attributes in the complete dataset.

File	Label	NB	Logist	MLP	SMO	IBk	AB	ONER	PART	J48	RF
All files	BENIGN	0.831	0.985	0.875	0.975	0.998	0.982	0.993	0.998	0.998	0.998
	ATTACK	0.663	0.980	? = 0	0.962	0.992	0.936	0.975	0.993	0.993	0.992

Note: ? = 0 means F1 equivalent to 0, for those cases where TP = FP = 0.

**Table 20 sensors-22-09326-t020:** F1 performance results using Zeek over different labelling corrections in Friday file.

File	Label	PART	J48	RF
Friday (original)	BENIGN	0.998	0.998	0.997
	ATTACK	0.997	0.997	0.997
Friday (PS fixed)	BENIGN	0.999	0.999	0.999
	ATTACK	0.999	0.999	0.999
Friday (BOT fixed)	BENIGN	0.998	0.997	0.997
	ATTACK	0.997	0.997	0.997
Friday (PS and BOT fixed)	BENIGN	0.999	0.999	0.999
	ATTACK	0.999	0.999	0.999

**Table 21 sensors-22-09326-t021:** F1 performance results using Zeek over different labelling corrections in Wednesday file.

File	Label	PART	J48	RF
Wednesday (original)	BENIGN	0.999	0.980	0.999
	ATTACK	0.998	0.997	0.998
Wednesday (without Hulk)	BENIGN	0.999	0.998	0.999
	ATTACK	0.986	0.980	0.986

**Table 22 sensors-22-09326-t022:** Performance results of related works in the CICIDS2017 dataset.

Reference	ML Technique	F1 Score	FS Method	N Attr.	Comments
Sharafaldin et al. [25]	ID3	0.980	RandomForest Regressor	4 per class	Weighted evaluation metrics
Engelen et al. [48]	RF	0.990	NA *	NA *	
Rosay et al. [49]	RF	0.999	Not used	74	Dest. port included. Regeneration and labeling of flows
Abdulhammed et al. [50]	RF	0.988	PCA	10	IP addresses and ports included
Kurniabudi et al. [51]	RF	0.998	Information Gain ranker	15	Dest. port included
Meemongkolkiat et al. [52]	Bagging ensemble	0.999	Recursive Feature Elimination & RF	18	Dest. and source ports included. 30 percent of the dataset
Our work	PART/J48	0.999	Not used	77	
	RF/J48	0.990	CFS	6	
	PART/J48	0.997	Not used	14	Zeek-based flows and attributes

* NA = Not available.

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
