# Peer review of "Evaluation of Machine Learning Techniques for Traffic Flow-Based Intrusion Detection"

_sensors, 2022, doi:10.3390/s22239326_

Round 1

Reviewer 1 Report

The authors investigated the application and evaluation of classification ML techniques for distinguishing between traffic flows corresponding to attacks over CICIDS-2017.  The manuscript was well written (but can be improved) and the experimental process is adequate.

However, the following comments are suggested to improve the quality of the manuscript.

1. The introduction section should be extended to have a formal background of Network security and an explicit introduction to IDS and traffics flows. Statistics of current IDS application and traffic flows for network security can be included.

2. A paragraph can be included to explicitly highlight and summarize the major contribution to the body of knowledge/Novelty.

 3. The parameters of the implemented ML and FS methods should be included to aid the reproduction of the experimental process.

4. There should be references to the selection of the implemented ML and FS methods. For instance, existing such as

a. Ensemble and Deep-Learning Methods for Two-Class and Multi-Attack Anomaly Intrusion Detection: An Empirical Study

b. Ilyas, M.U. and Alharbi, S.A., 2022. Machine learning approaches to network intrusion detection for contemporary internet traffic. Computing104(5), pp.1061-1076.

c. Alshammari A, Aldribi A. Apply machine learning techniques to detect malicious network traffic in cloud computing. Journal of Big Data. 2021 Dec;8(1):1-24.

may be reviewed and its findings can be useful for the manuscript.

Also, FS studies such as 

a. Balogun AO, Basri S, Abdulkadir SJ, Hashim AS. Performance analysis of feature selection methods in software defect prediction: a search method approach. Applied Sciences. 2019 Jul 9;9(13):2764.

b. Balogun AO, Basri S, Mahamad S, Abdulkadir SJ, Almomani MA, Adeyemo VE, Al-Tashi Q, Mojeed HA, Imam AA, Bajeh AO. Impact of feature selection methods on the predictive performance of software defect prediction models: an extensive empirical study. Symmetry. 2020 Jul 9;12(7):1147.

c. Nguyen H, Franke K, Petrovic S. Improving effectiveness of intrusion detection by correlation feature selection. In2010 International conference on availability, reliability, and security 2010 Feb 15 (pp. 17-24). IEEE.

can be used to justify the use of CFS and BFS methods for attribute/feature selection.

5. Using only the F-1 score as the reported performance metric may not be adequate. Besides, other metrics such as accuracy recall, and precision were included in the manuscript but was not used. I will suggest you use Mathews Correlation Co-efficient (MCC) as a metric. It has been reported to be balanced and fair.

6. You may need to re-arrange the flow of the manuscript based on the template of the journal. The related works may need to come after the introduction section. This will be appropriate to justify the research problem and objectives with respect to existing studies.

7. A section on the threat to validity should be included in the manuscript. 

8. The findings from the studies in the related work section should be tabulated. This will make it easy for the readers to see the contribution of the research work in relation to existing studies.

9. The caption Discussion and Conclusions should be revised based on the template of the journal. Also, highlights on future works should be included.

Reviewer 2 Report

Some Suggestions for Improving the manuscript on Evaluation of machine learning techniques for traffic flow- 2 based intrusion detection are as follows:

1. Need to refine more in the Related Work/Literature Review Section. Proper comparison should be given.

2. Discussion and conclusion is not clear.

3. Each Table description is required, some tables are not proper.

4. Insufficient Literature Review.

5. Conclusions that do not appear to be supported by the study data.

6. Some Sentences are not clear and concise.

Reviewer 3 Report

1. Are your approach and analysed methods also applicable to data leakage protection (DLP) systems? Please comment on this in the paper.

2. You chose the CICIDS2017 dataset. Please clarify to what extent your results and conclusions are comparable/applicable when using other datasets.

3. It is quite unusual to have a "Related work" section near the end of the paper. I suggest you leave the content in which you compare your results with other's work, but rename the section name, e.g., "Comparison to related work", and try to be more precise in comparing (for example, show the comparison by using a table).

4. For tree-based techniques with CFS attribute selection eight attributes were selected. Please clarify why did you choose eight parameters? Compare the F1 results with cases when you would choose a slightly higher and lower attribute number, respectively.

5. In the "5. Discussion and conclusions" section statements like "not very long execution times", "requiring longer execution times" and "reduction in time was notable" should be more precisely formulated, e.g. with a view to possible real-time operation.

Round 2

Reviewer 1 Report

The authors have improved the quality of the manuscript by effecting the comments/corrections given by the reviewers. The manuscript should be accepted in its current state (subject to editorial corrections).